# The Economic Implications of Phasing Out Pig Tail Docking: A Pilot Study in Italy

**DOI:** 10.3390/ani15091250

**Published:** 2025-04-29

**Authors:** Francesca Menegon, Annalisa Scollo, Samuele Trestini, Rachele Urbani, Giuseppe Ru, Guido Di Martino

**Affiliations:** 1Istituto Zooprofilattico Sperimentale delle Venezie, Viale dell’Università 10, 35020 Padova, Italy; 2Department of Veterinary Sciences, University of Torino, Grugliasco, 10095 Torino, Italy; 3Department of Land, Environment, Agriculture and Forestry (TESAF), University of Padova, 35020 Padova, Italy; 4Epidemiology Unit, Istituto Zooprofilattico Sperimentale del Piemonte, Liguria e Valle D’Aosta, Via Bologna 148, 10154 Torino, Italy

**Keywords:** tail, docking, undocked, pig, costs, economic, tail biting, weaning, fattening, straw

## Abstract

The European Commission’s ban on routine tail docking has prompted the transition to undocked pigs, which promises improved welfare but entails uncertainties, particularly in costs. This retrospective observational study aims to assess the short-term effects of transitioning from routine tail docking to 100% undocked pigs in swine production, shedding light on productivity, health, and economic implications. Twenty-two farms were assessed during three subsequent phases: total tail docking (step 1), subgroups of undocked pigs (step 2), and fully undocked pigs (step 3). Farmers received training in long-tail management and independently implemented it on their own farms to improve pig welfare. However, straw provision as environmental enrichment was mandatory, at least supplied during periods of pigs’ restlessness. Overall, going through step 2 appears to be successful. However, transitioning to step 3 worsened mortality, tail lesions at slaughter, and feed conversion ratio, leading to higher costs for producing 1 kg of meat. The hypothetical labour required to optimize straw management, ensuring its continuous availability, was estimated to further increase expenses. The complexity of tail biting requires careful adaptation over time and increased management efforts to implement ideal management. Mandating only the non-continuous use of straw has proven insufficient.

## 1. Introduction

Tail biting (TB) represents a major welfare concern for commercial swine production on a global scale. This abnormal behaviour has a multifactorial origin, making this complex and multifaceted problem challenging to predict and control [1,2]. It can cause severe injuries and often displays an alarming exponential trend that rapidly involves a vast part of the farm population [3,4]. As a result, TB management impacts farm productivity and necessitates additional labour, thereby inducing notable economic consequences for farmers [5]. The difficulty in preventing TB outbreaks in intensive commercial farming conditions has led to the widespread adoption of tail docking as a preventive measure [3]. However, tail docking itself induces acute stress and pain [6], heightens pain sensitivity, and may lead to chronic pain [7,8], merely mitigating the repercussions of the issue without ensuring the eradication of tail biting episodes [9].

The European Commission (EC) undertook a strategic roadmap to address this welfare issue: it began with the prohibition of routine tail docking in 1994 via Directive 91/630/EEC [10], subsequently modified by Directive 2001/88/EC [11] and Directive 2001/93/EC [12], and repealed by Directive 2008/120/EC [13]. From 2009 to 2019, the EC audited measures to prevent TB and discourage routine tail docking, finding them inadequate and ineffective in the evaluated countries [14,15]. Civil society has significantly raised awareness [16], numerous scientific publications have investigated TB [2], and many recommendations have been made regarding the measures to reduce TB (2016/336) [17]. This trend is driven by the need to improve overall pig welfare, with tail biting recognized as a valuable iceberg indicator of underlying issues [18]. Furthermore, beyond the direct benefits to animal welfare, several studies have highlighted a strong association between improved welfare conditions and enhanced farm productivity parameters [19,20].

The situation is very heterogeneous in Europe due to stricter national rules than EU legislation [21]. Tail docking is no longer allowed in Finland and Sweden, while in Norway and Switzerland, less than 5% of pigs are tail docked [3,21]. In contrast, official audits conducted between 2016 and 2018 in the leading European pig-producing nations, including Germany, the Netherlands, Italy, Spain, and Denmark, highlighted the widespread prevalence of tail docking, with 95% to 100% of pigs subjected to the practice [21,22]. The European Commission urged these countries to provide national inspectors with instructions and guidance to enable them to apply provisions for TB prevention and avoidance of routine tail docking. Finally, the official veterinarians were encouraged to assess the incidence of TB on farms and the effectiveness of implemented improvement measures [16].

All these countries adopted action plans to gradually transition through the prevention of TB and avoidance of tail docking. The “Action Plan to Revitalise the Pig Production Sector” was proposed in the Netherlands [23], and the Danish Veterinary and Food Administration (DVFA) stated that tail docking will only be allowed if the farmer has written documentation that efforts are being made to reduce TB, as in Germany [24]. Italy published an Action Plan in 2018 [25] addressing the recommendations of the Food Veterinary Office (FVO) in terms of risk analysis. Moreover, pig producers must introduce a group of undocked pigs into their farms, ensuring the effectiveness of implemented measures and gradually increasing the number of undocked pigs (Ministerial note prot. n. 27,719 of 2 November 2023). The FVO further suggested monitoring tail lesions. Consequently, the European Commission endorsed funding for a system to automatically detect tail length and severity of lesions at slaughterhouses, providing crucial insights to improve farm management practices [26].

Data regarding the additional costs and economic losses incurred during the transitional phase within the same farms remain scarce. The 100% undocked weaner or fattening pigs process is complex and challenging [27]. Many risk factors can play a very different and unexpected role, making this transition extremely tortuous. Its complexity requires a dynamic management adjustment strategy approach to address the limitless variety of intermediate stages [3].

This study aims to assess the short-term effects of transitioning from routine tail docking to 100% undocked pigs in swine production, shedding light on productivity, health, and economic implications. The study retrospectively evaluated three subsequent phases concerning tail length, namely, 100% docked pigs, 10–50% undocked pigs, and 100% undocked pigs, in a sample of weaning and fattening farms that underwent the phased elimination of tail docking.

## 2. Materials and Methods

The data were collected retrospectively from 4 weaning and 18 fattening intensive commercial farms (Table 1) applying all-in/all-out procedures. All farms were located in Emilia-Romagna and Lombardy, two Italian regions highly populated by intensive pig farms, and were designated for the production of heavy pigs (over 160 kg b.w. at slaughter). As the duration of the weaning or fattening phases is different (about 8–11 weeks and 6 months, respectively), the project considered weaning batches from 2019 to 2022 and fattening batches from 2015 to 2022 to ensure a proper sample size. All the farms (*n* = 22) included in the study were involved in convenience sampling. They were randomly selected from a list provided by a contractor under a farming agreement, as suggested by de Oliveira Sidinei et al. [28]. This selection procedure was adopted to minimize the effect of contractual requirements and different types of contractor–farmer relationships. From the list of pig farmers under the same contractual agreement, the farmers involved in the study voluntarily incentivized the elimination of tail docking. In total, 36 batches of weaning piglets and 84 batches of fattening pigs were involved in the study.

Data on the 36 batches of weaning piglets originated from two Italian farrowing farms organized with a 3-week batch system. The batch was composed of piglets weaned at the same farrowing site on the same day. The weaning pigs’ genetics were Hypor x Fomeva, and all the pigs were fed ad libitum with dry feed in automatic feeders. Male piglets were routinely castrated in farrowing farms within seven days of age, in accordance with European Law requirements [14], and voluntarily using analgesia. At the same time of castration, the piglets selected for docking were subjected to electrocautery for tail removal. On arrival at the weaning site, the pigs were housed according to the stocking density required by Council Directive 2008/120/EC for 7–10 kg of body weight (0.15 m^2^/animal) [13]. Then, at 20 kg, they were split into smaller groups to guarantee the legal stocking density (0.30 m^2^/animal). They were reared on a partially slatted floor, and microclimate conditions were maintained by controlled mechanical ventilation. In each pen, drinking water was provided ad libitum via at least one nipple drinker; based on pen sizes across farms (minimum = 8; maximum = 80 pigs), the drinker-to-pig ratio was consistently maintained at no less than 1:20. The same feed factory supplied all the weaning farms.

Regarding fattening, 84 batches were included in the study. A batch was defined as a group of animals of the same age that was moved on the same day to the study farm; 42 originated from Denmark, 2 from the Netherlands, while the animals of the remaining 40 batches were born in Italy and came from the 4 Italian weaning farms (named “a” to “d” in Table 1) involved in this study. Italian and foreign pigs were never mixed in the same batch. However, Italian fattening batches may consist of weaned pigs from more than one weaning farm, depending on farm size. No information was available about the weaning farms from which the Danish and Dutch fattening pigs originated. The genetic was Hypor x Fomeva for the Italian pigs and unknown for the Danish and Dutch ones. Pigs were fed liquid feed twice or three times a day depending on the farm; feed distribution was made on long troughs that allowed all the animals to eat simultaneously (at least 40 cm/pig of individual space at the trough). All the fattening farms were supplied by the same feed factory. In each pen, drinking water was provided ad libitum via at least one nipple drinker; based on pen sizes across farms (minimum = 13; maximum = 28 pigs), the drinker-to-pig ratio was consistently maintained at no less than 1:14 for fatteners. All the fattening pigs in this study were destined for heavy pig production (slaughtered at nine months of age and 170 kg of live weight). Italian pigs were also designed for the Protected Designation of Origin (PDO) ham production.

According to tail length/management, every batch was classified as follows:Step 1: 100% of the pigs on the farm were tail docked.Step 2: Ten to fifty percent of the pigs on the farm had undocked tails; they were reared in mixed groups with docked ones. At weaning, undocked litters arrived already mixed with docked ones. From that point onward, undocked pigs were raised together with docked pigs. For logistical and hierarchy-related reasons, they were not separated after being mixed.Step 3: 100% of the pigs on the farm had undocked tails.

Table 1 outlines key details for each farm. The number of batches observed per year is given in Appendix A.

Farmers attended a training course on managing undocked pigs in the two months before advancing from step 1 to subsequent phases. This course, organized by a private company specialized in the field and repeated for each farm (or small group of farms, when possible), provided an overview of relevant risk factors and proposed potential solutions. Each farmer independently decided which risk factors to address on their farm, but the presence of straw as environmental enrichment was made mandatory across all farms. Consequently, farms moving to steps 2 and 3 equipped each pen with a metal rack for straw (large: 39 L; small: 23 L; 5 × 5 cm^2^ mesh). In response to the farmers’ need to ensure the practicality of the intervention in large-scale commercial farms, straw was made mandatory at least when early signs of restlessness were observed (e.g., hanging tails, tail-in-mouth behaviour [29], increased pig vocalizations [30]) during daily farmers’ inspections of the animals, which are compulsory according to Dir. 1998/58/EC. All the racks were hung over a solid floor area that allowed the pigs to root the little straw sticking out from the meshes. In farms with fully slatted floors (namely farms “e, g” and “s” in Table 1), racks were hung over the trough to allow rooting behaviour. Some farms also provided pens with manipulable chains, plastic toys, or oak wood stumps (20 cm in length and 8 cm in diameter) hanging from the wall. The other risk factors considered during the training were customized and managed individually by each farm based on their specific weaknesses, leading to varied improvements across the farms. A 30% reduction in stocking density of Dir. 2008/120/EC was suggested as a key risk factor for improvement. The farm veterinarian supported farmers as needed and verified the actual implementation of the plan. An overview of the risk factor mitigation strategies implemented during steps 2 and 3 of the study are given per each farm in Appendix A.

### 2.1. Data Collection

For each batch, data were collected regarding the tail step, average weight at the beginning and the end of the growing cycle, average days on the farm, mortality (%), age in days at the time of pig death, and feed consumption. Using these data, the feed conversion ratio (FCR), feed yield (%), average daily gain (ADG = (final weight − initial weight)/average days on farm), and average weight gain (AWG = (final weight − initial weight)/n. pigs purchased) were calculated.

#### 2.1.1. Economic Data

The economic data collected included the cost of feed, drugs, animal purchases, metal racks purchased for straw, contractor fees, and total cost per kg of meat produced. To avoid market fluctuations unrelated to the study’s aim, the costs of feed, drugs, and animals were standardized based on the prices of the last year considered (2022). The characteristics of the feed used did not vary over the years, and, at the time of the study, the piglet market did not vary depending on the tail length. The feed cost per 100 pigs produced was also estimated. Contractor fees included all the expenses related to personnel and the use of buildings and farm structures; the total cost also included other costs not detailed (e.g., depreciation and electricity bills) and costs incurred for disposable environmental enrichments (straw, wooden logs, chains, plastic toys). The contractor covered the veterinary fees, flat-rated by the number of raised pigs per year for all the farms and included in the total cost.

#### 2.1.2. Drugs Use

Data on administered antimicrobials and anti-inflammatory (nonsteroidal anti-inflammatory drugs—NSAIDs—and corticosteroids) drugs were obtained from the treatment registries. The recorded product consumption was converted into the active substance used in milligrams, which was then transformed into the Defined Daily Dose per animal (DDDita) based on the Italian Ministry of Health Guidelines [31,32]. The number of DDDita was divided by the number of animals produced in each batch, and by the expected weight at treatment, which for Italian heavy pigs was proposed to be 100 kg of live weight during fattening and 12 kg of live weight during the weaning phase, in agreement with previous studies [33,34]. The maximum value was chosen when the daily dosage was within a range [33,35].

#### 2.1.3. Tail Lesions at Slaughter

At the end of the fattening cycle, an expert veterinarian (i.e., the second author of the present study, who remained the same throughout the study) assessed tail lesions by applying a 2-point scale protocol on carcasses at the slaughterhouse: score 0 = no lesions; score 1 = from mild lesions, redness, irritation, scratches, or minor abrasions to severe lesions with bleeding and loss of tissue (modified from Di Martino et al. [36]). A binary score was preferred due to the high likelihood of healed lesions being detectable at slaughter in heavy pigs [37] and the absence of a thoroughly reliable and suitable method to visually identify in the slaughter line when a lesion has occurred along the pig’s growing cycle [38,39]. Tail assessment was visually made after scalding and dehairing of carcasses, to increase lesion detectability [40,41]. The evaluation was conducted from a one-metre distance, directly under a light source.

#### 2.1.4. Clinical Scores at Weaning

Additional data were collected for Italian batches at the farrowing site regarding the prevalence of some clinical syndromes during lactation: enteric, cutaneous, neurologic, and locomotor scores (0–4) were assigned based on the prevalence of symptoms observed in the entire batch, as reported in Table 2. The prevalence of each clinical syndrome was derived from daily health records maintained by expert technicians at the farrowing farms. Clinical scores were assigned early in life to identify potential health disparities in pigs subsequently involved in the study and to mitigate potential bias related to mortality during the weaning and fattening periods caused by prior health conditions.

#### 2.1.5. Estimation for Labour Costs Related to Ideal Straw Management

As the real straw consumption was not available by batch during the study, the straw cost was included in the total cost, and expenses related to personnel and workers’ labour related to straw management were included in contractor fees. Based on these assumptions, labour costs associated with optimal straw management were estimated. Based on the literature, ideal straw management involves providing animals with continuous and ad libitum access [3]. The ideal daily straw consumption considered for the estimation was 5 g/pig/day during the weaning phase [36], and 70 g/day/pig during the fattening cycle [43], including possible losses of straw due to the partially slatted floor. These amounts were chosen because previous studies on this specific livestock production found these daily consumption rates during weaning and fattening. The total amount of straw ideally required by the entire herd during each productive phase was calculated by multiplying the pig daily straw need (g)/pig/day × average days on farm × batch size (number of pigs). The result was divided by the volume of straw racks present at the farm (big 39.00 L, small 23.00 L) converted into grams (straw specific weight 0.05 g/cm^3^) [44] multiplied by the total number of racks. The output indicated how many times racks should be filled to allow continuous straw availability for the entire productive cycle. To estimate the costs related to the labour needed for hypothetical straw distribution, the farmers’ time spent on straw distribution was collected in a subsample of 12 farms (3 weaning and 9 fattening farms; see Table 1) and multiplied by straw deliveries. The 12 farms were visited, and a researcher timed the straw distribution on the entire farm using a chronometer. Straw distribution was always manually performed by one stockman. The time for all procedures involved in straw distribution (e.g., moving the straw from storage to the barn, operator breaks) was included. For the remaining non-visited farms, the distribution time was estimated by calculating the mean distribution time per rack multiplied by the number of racks on the farm. The number of hours that resulted in straw distribution during the entire weaning or fattening cycle was standardized to 100 animals. The hours one person ideally spent distributing straw was multiplied by the average gross hourly pay of farm workers per hour with two years of experience in the Lombardy Region in 2022 (12.60 EUR/h [45]). This area can be considered a single productive contest from a salary point of view. Finally, the average cost of straw distribution for 100 pigs was estimated for a weaning and a fattening cycle.

### 2.2. Statistical Analysis

All variables were found to have a normal distribution. An ANOVA was conducted to investigate statistically significant associations with production, economic, and health parameters among the three tail steps. Subsequently, a post hoc test was performed using Bonferroni adjustment to identify which pairs of groups significantly differed. Differences between means with *p* < 0.05 were accepted as statistically significant. Significance levels of *p* < 0.10 were still discussed.

Moreover, the influence of mortality on costs to produce one kilogram of meat (for feed, pigs purchasing, drugs, contractor, feed per 100 pigs, and total cost) and feed yield was estimated. The mortality (independent variable) effect on the other factors (dependent variables) was calculated using a linear regression model. The analyses were conducted using IBM SPSS (version 26.0).

## 3. Results

### 3.1. Weaning

The study involved 52,370 weaned piglets. The batch sizes ranged from 615 to 1970 pigs, averaging 1455 piglets per batch. Concerning the percentage of undocked pigs, 16 batches were step 1, 8 were step 2, and 12 were step 3. Feeding 100 pigs during the weaning phase costs EUR 2976.90 ± 395.50.

When a group of undocked pigs was introduced on the farm (step 2), the production parameters remained unchanged compared to step 1 (Table 3). Conversely, transitioning from step 2 to raising the entire population with intact tails (step 3) deteriorated production parameters such as mortality, FCR, and feed yield (*p* = 0.010, *p* = 0.015, *p* = 0.025, respectively). At the farrowing site, a lower weaning weight was recorded for step 3 than for steps 1 and 2 (*p* < 0.001).

Concerning costs, step 3 was associated with a statistically significant increase in expenses for feed, piglet purchasing, and total cost to produce one kg of meat compared to step 2 (*p* = 0.019, *p* = 0.049, *p* = 0.039, respectively). Contractor fees were higher in step 3 than in step 1 (*p* = 0.014) and step 2 (*p* = 0.044). Overall, producing one kilogram of meat in step 3 costs 33.9% more than in step 1 and 47.9% more than in step 2 (specifically, with a feed cost of +32.5%, a piglet purchasing cost of +53.6%, and contractor fees of +76.3%). Regarding clinical data, step 3 involved animals showing a greater incidence of cutaneous syndrome (*p* = 0.008) during lactation. More drugs were used than in step 2 (*p* = 0.017) while drug costs remained unchanged.

No other differences were found.

### 3.2. Fattening

The study included 167,607 fattening pigs. The batch sizes ranged from 340 to 6358 pigs, with an average of 2019 pigs. There were 65 batches in step 1, 7 in step 2, and 12 in step 3. Feeding 100 pigs during the fattening phase costs EUR 12,625.45 ± 2001.33. All the non-Italian pigs fell into step 1.

The entry and slaughter weights of step 1 were lower than those of the step 2 pigs (*p* = 0.068, *p* = 0.012), and the slaughter weight and number of days on the farm were lower than those of the step 3 pigs (*p* = 0.013, *p* = 0.009). Step 1 showed improved FCR and feed yield compared to steps 2 and 3 (FCR: *p* = 0.037, *p* < 0.001; feed yield: *p* = 0.028, *p* < 0.001), higher ADG, and lower mortality than step 3 (*p* = 0.004, *p* = 0.002). The FCR and feed yield did not significantly differ between step 2 and step 3 (Table 4).

The introduction of a group of undocked pigs (step 2) was associated with increased feed cost (+8.35%), and cost to feed 100 pigs (+18.20%) compared to step 1 but lower cost of drugs (−53.85%) and cost of piglet purchasing (−12.32%) to produce one kg of meat. Raising undocked pigs (step 3) resulted in increased feed cost (+10.90%), contractor fees (+13.21%), total cost, and cost to feed 100 pigs (+18.87%) compared to step 1. Producing 1 kg of meat in step 3 batches costs 7.38% more than in step 1 and 8.44% more than in step 2.

The prevalence of tail lesions at slaughter was significantly greater in step 3 batches (41%), followed by step 2 (10%) and step 1 (1%) (Table 4). Besides the lower cost of drugs, step 2 used 10.6% more DDDita than step 1.

### 3.3. Other Costs, Labour Estimation for Straw Management, and Economic Influence of Mortality

The price of the small rack was EUR 55, while the price of the large rack was EUR 70. Assuming that the expense for racks is spread over ten years, the per pig cost can be deemed insignificant.

Distributing the ideal amount of straw to 100 pigs during a weaning cycle was estimated to take an average of 35 min, and the estimated cost was EUR 4.37. For a heavy pig fattening cycle, the average time was estimated at 10.5 h, with a cost of EUR 109. Distributing straw in small feeders was estimated to take three times as long (an average of 52 min per weaning cycle and 26.5 h per fattening cycle) compared to using large feeders (an average of 17 min per weaning cycle and 7.3 h per fattening cycle).

Regarding the economic impact of mortality, observations indicate that the median mortality occurred around the 40th day of the weaning cycle and the 74th day of the fattening cycle, the halfway point in both cases. Consequently, upon an animal’s death, 100% of the purchasing cost, 100% of contractor fees, and presumably 50% of the feed cost are lost. Data on the date of death were collected for 3551 dead piglets in the weaning phase and for 7312 dead fattening pigs.

Regressing various economic indices on the mortality rate (Appendix A) revealed a statistically significant association between mortality and all analyzed costs at weaning, except for “feed cost per 100 piglets”. Specifically, for each unit increase in mortality percentage, the feed cost, piglet purchasing cost, drug cost, contractor fees, and total cost per kg of produced meat increased by EUR 0.03, EUR 0.11, EUR 0.01, EUR 0.02, and EUR 0.17 per kg of life weight gain, respectively. The feed yield decreases by 0.33% for every 1% increase in mortality. During the fattening phase, the impact of 1% increase in mortality on production costs of produced meat remained evident, with an increase in cost per kg of live weight gain equal to EUR 0.01 for feed, EUR 0.01 for pigs purchased, EUR 0.03 for contractor fees, EUR 0.03 for total cost. Consequently feed costs increase by EUR 220 per 100 pigs while feed yield decreases by 0.47% for every 1% increase in mortality.

## 4. Discussion

The whole point of the EU legislation is to ensure that, since pigs are legally recognized as sentient beings, there must be changes in current intensive pig husbandry. Indeed, tail biting is triggered by frustration and an inappropriate environment, to which tail docking does not represent the solution. Instead, it is more likely an inexpensive subterfuge to the issue from an ethical point of view.

This is the first study, to the authors’ knowledge, considering the short-term effects of transitioning from routine tail docking to 100% undocked pigs on the same farms. This study was not conducted under a controlled experimental setting, but it utilized retrospective data collected from intensive commercial farms. This approach enables observing the effect of the various steps within a real-world context characterized by the specificity of the pig production sector and the uniqueness of individual farms. Participants opted into the tail docking elimination program voluntarily; consequently, the recruited farmers were likely motivated to some extent. The results from weaning sites, which represent the highest risk for TB due to the age of the animals [46], indicate that rearing a group of undocked pigs among the docked batch (step 2) appears to be a safe transition towards abandoning tail docking in terms of productive parameters. Conversely, moving from step 2 to step 3 under the present study’s conditions had a statistically significant negative impact, with greater mortality and detrimental FCR and feed yield.

A plausible interpretation of this study’s results is that the farmer’s alertness increases when a group of undocked pigs is first introduced, but the risk of widespread damage remains limited. However, when raising an entire group of undocked pigs, the risk escalates exponentially, requiring efforts to be spread across a larger population. Even under favourable conditions, TB may still occur due to its multifactorial nature. Although many risk factors are well known [4], identifying the actual causes of TB in each farm is often complex, as the combined effect of many factors triggers sporadic and unpredictable outbreaks. Among protective factors, the efficacy of straw as environmental enrichment is well documented [4]. In the present study, straw was introduced as part of a compromise with farmers, mandatory at least when the animals exhibited increased activity, as a strategy to reduce the workload associated with large-scale straw management. However, this intermittent availability itself might constitute a risk factor for TB [4]. Furthermore, no additional mandatory measures were consistently implemented across farms (e.g., improved resource availability such as water flow rate, enhanced control of infectious diseases, or increased air quality management), even though, on average, farms appeared to adopt a lower stocking density than that outlined in current EU legislation (0.51 vs. 0.30 m^2^/pig at the weaning sites, and 1.33 vs. 1.00 m^2^/pig at the fattening sites). This open approach, which mainly relies on the highly variable and subjective managerial decisions of farmers and their ability to promptly identify early signs of restlessness across all pens before introducing straw, may contribute to an increased risk of failing to transition to step 3 safely. Based on weaning data, a slight influence of the increased incidence of cutaneous clinical syndrome during lactation on the worsened FCR and feed yield in step 3 cannot be excluded. However, this is likely not directly related to tail length. The authors suggest this influence is barely relevant, as an average cutaneous score of 1.75 (as seen in step 3) indicates a cutaneous syndrome prevalence of less than 10% in lactating piglets. Similarly, the connection between worsened nutritional parameters and a slight increase in the incidence of enteric clinical syndrome cannot be excluded [47]. However, its relevance seems minimal, as it is supported only by a statistical tendency.

In the fattening phase, step 3 registered increased mortality and decreased FCR, feed yield, and ADG compared to step 1. Moreover, a 40-fold greater prevalence of tail lesions was observed at slaughter in step 3 than in step 1, confirming the findings of other authors reporting that tail length represents a risk factor for tail lesions [46,48,49]. This result might partially explain the reduced efficiency of the productive parameters. Van Staaveren et al. [38] estimated that even a prevalence lower than 1% of severe tail lesions in docked batches is associated with a considerable decrease in ADG (−4.8%), translating into pigs requiring 7 days more to reach target slaughter weight. This meant that feed costs increased by 1.5% per year, and the mean annual farm profit was reduced by 15.1% in farms with a higher prevalence of severe tail lesions [38]. However, steps 2 and 3 were also associated with a prolonged rearing period before slaughter and were used to reach a greater slaughtering weight than step 1 due to PDO requirements. These differences are likely linked to the inclusion of step 1 batches originating from foreign countries (while in steps 2 and 3, they were 100% Italian), therefore, not intended for DPO production even if raised for heavy pig production. The absence of a mandatory threshold for age and weight for Italian DPO pigs (at least 9 months and 165 kg, respectively) might lead to a different slaughtering time (192.9 vs. 170.5 days on the farm) and weight at slaughter (174.5 vs. 162.3 kg of body weight). This further explains the worsened FCR and feed yield of steps 2 and 3 compared to step 1 as, considering the standard growth curve of a pig, increasing days on the farm corresponds to a slow deterioration in production indices [50].

When productive data are translated into economic data, according to our results, producing one kg of meat from undocked pigs (step 3) costs 33.9% more during the weaning phase (5.00 compared to 3.74 EUR/kg produced) and 7.38% more during the fattening phase compared to step 1 (1.64 compared to 1.53 EUR/kg produced). This mainly reflects the higher costs for feed because of the reduced feed yield, losses of kilograms of meat due to mortality during the cycle, and contractor fees due to the different economic agreements for personnel remuneration that occurred after undocked animals were introduced. Mortality seems to have a relevant impact on the economic sustainability of step 3, considering that each +1% increase is equivalent to a EUR 0.02 increase in total cost per kg of live weight produced at the weaning site and EUR 0.03 at the fattening site. In contrast, the authors suggest an irrelevant influence of tail docking on the cost of piglet purchasing, as it was normalized to the market price in 2022 to avoid market fluctuations. The significant difference shown by the results is probably linked to the different weights of the animals at the beginning of the cycles, which may not depend on tail length. The probably consequent tendency to gain less weight during weaning (*p* = 0.060) might also lead to a lower weight at the beginning of the fattening cycle and to a higher cost per kg of the single animal purchased (the greater the weight, the lower the cost per kg).

Moreover, the total cost included husbandry adaptation costs, which strongly depended on the farm’s baseline situation regarding equipment and management. During step 3, farmers could have implemented large-scale strategies to handle enrichments (e.g., a quarantine deposit for straw), additional consulting with an animal welfare expert, and investments in environmental condition monitoring and settings. In addition, costs related to the possible increase in animals undergoing euthanasia due to bite-related damage should be considered (they are also included in the total cost). According to Hoste et al. [51], in farms that have already invested in beyond-legal animal welfare measures, costs may amount to about EUR 9 per delivered pig, of which EUR 4 are for the piglet phase. In the case of poorly modernized farms, which might apply to traditional Italian farms—such as those in our sample at the time of their involvement in the study [52]—costs might increase by up to 188% (EUR 26 per delivered pig, of which EUR 10–11 is during piglet production). Moreover, additional costs must be considered during the farmers’ learning phase (2–5 years). These costs amount to approximately EUR 3 per delivered pig, representing an increase of 11.5% [51].

The present study included labour costs related to straw management in contractor fees. Nevertheless, the time required to manage this environmental enrichment optimally was estimated based on permanent access, as proposed in the literature [3], as an effective measure to prevent and reduce the risk of TB and other pathological conditions (gastric ulcers) [43,53]. Indeed, the negative outcomes associated with the transition to step 3 observed in this study do not appear to support the strategy of making the continuous provision of straw discretionary. The estimated amount of labour needed to guarantee a correct environmental enrichment was 35 min/100 piglets per weaning productive cycle (EUR 4.37) and 10.5 h/100 pigs per fattening cycle (EUR 109) in case the metal racks are easy to fulfil. The estimate would also increase following the recommendations of other authors focusing on gastric ulcers, who proposed an optimal straw distribution ensuring ad libitum access during the fattening cycle of more than 500 g/pig/day [53]. Indeed, the considerable amount of time estimated for proper environmental enrichment management might be perceived as a limitation in the field.

Concerning drug consumption and costs, step 3 was comparable to step 1. However, beginning in May 2022, zinc-oxide treatments (e.g., in the last semester of the study, step 3) were forbidden [54]. The ban on zinc treatment could have induced a general increase in the use of antimicrobials at weaning sites to replace it, even though there is a decreasing global trend in antimicrobial use in veterinary medicine [55]. Treatments containing zinc were not included in the quantification of DDDita, following the guidelines supplied by the Italian Ministry of Health [33].

Costs related to carcass condemnation after slaughtering, which are reported to increase in the case of tail abscesses [49,53,56], were not available. It was estimated that a loss of 43% of the profit margin per pig is due to carcass trimming (EUR 1.10 per pig) and carcass weight reduction related to tail lesions (EUR 0.59) [56]. These estimates regard docked pigs; we can assume that profit losses on undocked batches may be more significant due to higher tail lesion incidence [48]. As data on abscessation and osteomyelitis are related to higher costs, it would be helpful to include them in further investigations [57].

This study has some limitations that warrant consideration. (I) The analyzed sample was relatively limited in size. Collecting a larger and more homogeneous sample would be desirable to ensure greater representativeness of the reference population. Additionally, (II) the effect of mortality on costs was calculated by assuming a linear growth curve instead of a standard growth curve (the older the dead pig was, the greater the cost), potentially underestimating the results. (III) The effect of an imperfectly balanced slaughter weight among steps might have partially affected feed efficiency in step 3. (IV) Detailed data regarding the causes of mortality are unavailable due to the study’s retrospective nature, which precludes the investigation of a possible relationship between tail docking and health issues, aside from the initial clinical classification of batches post-lactation. (V) Collecting data during a heavy pig production cycle required an extended observation period. Despite the distribution of the steps over more than a year, the “year effect” cannot be completely ruled out. (VI) Regarding weaning data, it is impossible to exclude the influence of the increased incidence of cutaneous syndrome during lactation on the worsened FCR and feed yield during step 3. Instead, it might be helpful to know whether the variability of these factors, such as litter unevenness, is also a risk factor for TB [58]. Moreover, (VII) collecting tail lesions while distinguishing between mild and severe cases might provide further valuable insights into the issue.

## 5. Conclusions

This study highlights strong ethical concerns regarding an effective resignation process of tail docking. Once the practice is banned, farmers will have to change many aspects of the pigs’ environment to maintain economic sustainability. Under the conditions of this study, rearing a limited group of undocked pigs was successfully implemented, whereas transitioning to a fully undocked group was not. Mandating only the non-continuous use of straw has proven insufficient, and greater efforts must be systematically implemented, even though it represents a more acceptable strategy for farmers due to the reduced workload. Further investments appear essential to consider before progressing to step 3. In the short term, keeping 10–50% of undocked pigs could help farmers identify management and structural aspects needing modification or improvement. The complexity of TB involves numerous risk factors, and adaptation requires time and gradual changes, especially at the beginning of this transition.

## Figures and Tables

**Table 1 animals-15-01250-t001:** An overview of the farms involved in the study, identified by letters (a–v): whether the farm was visited by the operator during the data collection process, the type of environmental enrichment implemented during steps 2 and 3 of the study, the tail management step completed by the farm (tail step: 1 = 100% docked pigs; 2 = 10–50% undocked pigs; 3 = 100% undocked pigs), the average batch size (number of animals of the same age present on the farm), the stocking density implemented during steps 2 and 3, the type of flooring (0 = fully slatted floor; 1 = 70% slatted, 30% concrete; 2 = 70% concrete, 30% slatted; 3 = fully concrete), and the volume of the racks for straw supply.

Farm	Visited	Environmental Enrichments	Tail Step	N. Animals/Batch	Stocking Density (m^2^/pig)	Floor Type	Racks’ Volume (L)
Weaning							
a	no	Straw, chain	1–2–3	1983	0.54	1	23
b	yes	Straw	1–2–3	1796	0.43	1	39
c	yes	Straw	1–2–3	2171	0.50	2	39
d	yes	Straw	3	1195	0.62	2	23
Fattening							
e	yes	Straw	1–2–3	1900	1.15	0–2	39
f	yes	Straw	2	1400	1.33	2	39
g	no	Straw, plastic toys	1–2–3	3066	1.03	0–2	39
h	yes	Log, chain	1	2320	1.04	0	-
i	yes	Straw	1–2–3	1530	1.35	2	23
j	yes	Straw	1–2–3	1109	1.35	2	39
k	yes	Straw	1–3	348	1.31	2	39
l	no	Straw	1–2–3	284	1.37	2	39
m	yes	Straw	1	913	1.31	2	39
n	no	Straw	1–3	662	1.35	2	39
o	no	Straw	1–2	626	1.35	2–3	39
p	no	Log, chain	1	1751	1.11	2	-
q	yes	Straw	1–3	3230	1.03	2	39
r	no	Log, chain	1	5888	1.01	2	-
s	yes	Straw, chain	1–2–3	1970	1.39	0	39
t	no	Log, chain	1	2750	1.06	0	-
u	no	Log, chain	1	780	1.05	2	-
v	no	Log, chain	1	3397	1.04	0–2	-

**Table 2 animals-15-01250-t002:** Scoring system based on clinical symptoms prevalence at batch level observed during lactation phase.

Score	Description	Prevalence				
		0	1	2	3	4
Enteric	Piglets with profuse yellowish diarrhea. Bacteriology on rectal samples isolated *Escherichia coli*. Colonies surrounded by a zone of lysis after overnight growth at 37 °C on blood agar were classified as hemolytic, and detection of virulence factor genes F18, STa, and STb was obtained by PCR.	≤5	>5; ≤10	>10; ≤20	>20; ≤30	>30
Cutaneous	Piglets with exudative epidermitis. Bacteriology from skin wounds or pus (cultured on MacConkey’s Agar) allowed to isolate *Staphylococcus* spp. The identification of *S. hyicus* was obtained by PCR (diagnosis: exudative epidermitis or “Greasy pig disease”) [42].	≤2.5	>2.5; ≤5	>5; ≤10	>10; ≤18	>18
Neurologic	Piglets with neurological signs. Bacteriology from the brain (cultured on blood agar supplemented with NAD and Gassner agar) allowed to isolate *Streptococcus suis*.	≤0.3	<0.3; ≤0.6	>0.6; ≤1.2	>1.2; ≤1.7	>1.7
Locomotor	Piglets with at least one leg with signs of inflammation (*calor, rubor, tumor, dolor, functio laesa*) without traumas. Bacteriology from intra-articular fluid (cultured on MacConkey’s Agar) allowed to isolate *Staphylococcus* spp. The identification of *S. hyicus* was obtained by PCR.	<0.5	>0.5; ≤1.5	>1.5; ≤3.5	>3.5; ≤4.5	>4.5

**Table 3 animals-15-01250-t003:** Data of three management models characterized by different percentages of weaned pigs with intact tails. Tail steps: 1 = 100% docked pigs (number of batches examined = 16); 2 = 10–50% undocked pigs (number of batches examined = 8); 3 = 100% undocked pigs (number of batches examined = 12). FCR = feed conversion ratio; ADG = average daily gain.

	Step 1	Step 2	Step 3		
	Mean ± St. Dev.	Mean ± St. Dev.	Mean ± St. Dev.	F	*p*-Value
Productive data					
Initial weight/pig (kg)	7.71 ± 0.34 ^a^	7.74 ± 0.52 ^a^	6.54 ± 0.50 ^b^	29.09	<0.001
Final weight/pig (kg)	36.09 ± 4.53	35.77 ± 2.72	33.22 ± 2.99	2.21	0.126
Days on farm	73.25 ± 3.89	72.00 ± 3.30	74.58 ± 3.20	1.30	0.285
Weight gain (kg)	26.10 ± 5.34	26.78 ± 2.62	22.04 ± 5.47	3.07	0.060
Stocking density (m^2^/pig)	0.51 ± 0.11	0.53 ± 0.09	0.51 ± 0.12	0.37	0.692
Mortality (%)	6.56 ± 4.93 ^b^	3.49 ± 1.25 ^b^	14.43 ± 11.77 ^a^	5.94	0.006
FCR	1.69 ± 0.09 ^ab^	1.56 ± 0.09 ^b^	1.73 ± 0.17 ^a^	4.68	0.016
Feed yield (%) ^1^	45.55 ± 2.97 ^ab^	49.63 ± 1.85 ^a^	44.59 ± 5.69 ^b^	4.25	0.023
ADG (kg)	0.38 ± 0.06	0.39 ± 0.04	0.34 ± 0.04	2.42	0.104
Economic data					
Cost of feed (€/kg produced) ^3^	1.20 ± 0.14 ^ab^	1.06 ± 0.05 ^b^	1.40 ± 0.41 ^a^	4.48	0.019
Cost of piglets purchasing (€/kg produced) ^2,3^	2.23 ± 0.62 ^ab^	2.01 ± 0.19 ^b^	3.09 ± 1.44 ^a^	4.13	0.025
Cost of drugs (€/kg produced) ^3^	0.08 ± 0.03	0.10 ± 0.04	0.12 ± 0.10	1.85	0.173
Contractor fees (€/kg produced) ^3^	0.23 ± 0.05 ^b^	0.22 ± 0.04 ^b^	0.40 ± 0.24 ^a^	5.44	0.009
Total cost (excluding VAT) (€/kg produced) ^3^	3.74 ± 0.78 ^ab^	3.38 ± 0.19 ^b^	5.01 ± 2.15 ^a^	4.40	0.020
Feed cost for 100 pigs (€)	3093.62 ± 475.03	2844.84 ± 331.97	2909.31 ± 287.01	1.34	0.275
Sanitary data					
Enteric score	1.38 ± 1.41	0.88 ± 0.84	2.17 ± 1.12	2.97	0.065
Cutaneous score	1.13 ± 1.26 ^ab^	0.13 ± 0.35 ^b^	1.75 ± 1.06 ^a^	5.67	0.008
Neurologic score	0.13 ± 0.34	0.00 ± 0.00	0.00 ± 0.00	1.31	0.284
Locomotor score	1.25 ± 1.57	2.13 ± 0.84	1.50 ± 0.67	1.44	0.251
Drugs consumption (DDDita/pig)	71,161 ± 24,173 ^ab^	44,701 ± 27,790 ^b^	84,935 ± 36,353 ^a^	4.43	0.020

^1^ Feed consumed (kg)/weight gain (kg) during the entire cycle. ^2^ The cost (€) per piglet is multiplied by the number of piglets purchased; the result is divided by the number of piglets exited × weight gain. ^3^ The cost (€) was divided by the weight gain (kg) increment. ^a,b^ Different letters mean statistical significance at *p* < 0.05.

**Table 4 animals-15-01250-t004:** Data of three management models characterized by different percentages of fattening pigs with intact tails. Tail steps: 1 = 100% docked pigs (number of batches examined = 65); 2 = 10–50% undocked pigs (number of batches examined = 7); 3 = 100% undocked pigs (number of batches examined = 12). FCR = feed conversion ratio; ADG = average daily gain.

	Step 1	Step 2	Step 3		
	Mean ± St. Dev.	Mean ± St. Dev.	Mean ± St. Dev.	F	*p*-Value
Productive data					
Initial weight/pig (kg)	30.05 ± 7.28 ^b^	36.22 ± 2.50 ^a^	33.59 ± 4.15 ^ab^	3.691	0.029
Final weight/pig (kg)	162.32 ± 14.71 ^b^	177.94 ± 4.59 ^a^	174.48 ± 4.38 ^a^	7.730	<0.001
Days on farm	170.46 ± 24.47 ^b^	183.43 ± 8.85 ^ab^	192.92 ± 21.42 ^a^	5.252	0.007
Weight gain (kg)	123.13 ± 14.69 ^b^	135.69 ± 5.19 ^a^	124.96 ± 5.44 ^ab^	2.837	0.064
Stocking density (m^2^/pig)	1.18 ± 0.24	1.08 ± 0.24	1.33 ± 0.50	2.079	0.132
Mortality (%)	5.60 ± 2.92 ^b^	3.39 ± 0.79 ^b^	9.06 ± 4.85 ^a^	8.555	<0.001
FCR	2.61 ± 0.2 ^b^	2.84 ± 0.20 ^a^	2.91 ± 0.17 ^a^	11.108	<0.001
Feed yield (%) ^1^	30.93 ± 0.03 ^a^	27.95 ± 0.02 ^b^	27.21 ± 0.02 ^b^	11.155	<0.001
ADG (kg)	0.80 ± 0.06 ^a^	0.77 ± 0.03 ^ab^	0.74 ± 0.04 ^b^	5.752	0.005
Economic data					
Cost of feed (€/kg produced) ^3^	0.74 ± 0.07 ^b^	0.81 ± 0.06 ^a^	0.82 ± 0.05 ^a^	10.846	<0.001
Cost of weaners purchasing (€/kg produced) ^2,3^	0.62 ± 0.08 ^a^	0.54 ± 0.02 ^b^	0.62 ± 0.07 ^a^	3.718	0.029
Cost of drugs (€/kg produced) ^3^	0.01 ± 0.01 ^a^	0.00 ± 0.00 ^b^	0.01 ± 0.01 ^ab^	3.316	0.041
Contractor fees (€/kg produced) ^3^	0.16 ± 0.02 ^b^	0.16 ± 0.01 ^ab^	0.18 ± 0.02 ^a^	4.695	0.012
Total cost (excluding VAT) (€/kg produced) ^3^	1.53 ± 0.20 ^b^	1.52 ± 0.07 ^b^	1.64 ± 0.12 ^a^	6.847	0.002
Feed cost for 100 pigs (€)	12,109.03 ± 1943.90 ^b^	14,313.79 ± 915.33 ^a^	14,394.82 ± 989.11 ^a^	11.753	<0.001
Sanitary data					
Enteric score	1.38 ± 1.22	1.00 ± 0.82	1.417 ± 0.93	0.397	0.675
Cutaneous score	0.77 ± 1.15	0.00 ± 0.00	0.90 ± 0.26	2.215	0.125
Neurologic score	0.29 ± 0.59	0.00 ± 0.00	0.54 ± 0.72	1.931	0.161
Locomotor score	1.68 ± 1.63	2.29 ± 0.49	1.38 ± 0.77	1.202	0.313
Drug consumption (DDDita/pig)	12.60 ± 8.92 ^b^	25.41 ± 17.63 ^a^	13.82 ± 9.11 ^ab^	3.038	0.058
Tail lesion score 0 (%)	98.91 ± 32.08 ^a^	90.00 ± 13.07 ^a^	58.98 ± 22.77 ^b^	6.747	0.003
Tail lesion score 1 (%)	0.97 ± 2.45 ^b^	10.00 ± 13.07 ^b^	41.02 ± 19.22 ^a^	32	<0.001

^1^ Feed consumed (kg)/weight gain (kg) during the entire cycle. ^2^ The cost (€) per piglet is multiplied by the number of piglets purchased; the result is divided for the number of piglets exited × weight gain. ^3^ The cost (€) was divided by the weight gain (kg) increment. ^a,b^ Different letters mean statistical significance at *p* < 0.05.

## Data Availability

Data are contained within the article or Appendix A.

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
