# Peer review of "The Economic Implications of Phasing Out Pig Tail Docking: A Pilot Study in Italy"

_animals, 2025, doi:10.3390/ani15091250_

Round 1

Reviewer 1 Report

Comments and Suggestions for Authors

Dear Authors

You raised an important issue. In my opinion the manuscript is clear and very interesing. The experimental desing is appropriate. The conclusions are presented well and align with the results obtained during the investigation.

Below you will find just a few comments to the text:

L81: What do you mean by the Competent Authority?

L114: Please add the information about their market weight.

L173: Please add the space after 0.

L225: Is it possible to provide readers with the photographs?

L255: Are you sure about this citation and the value of 70 grams a day in that context? The paper you cited refers to the gastroprotective effects of straw. Should we automatically assume that the value represents the ideal daily straw needs?

L384: Isn't is Authors'?

L501: Please be consistent with the punctuation: e.g., (see L473).

L709: 2022 (bold), 122 (not).

L711: 2020 (bold).

Best regards

Author Response

L81: What do you mean by the Competent Authority?

Authors: we have rephrased more clearly “official veterinarians”.

L114: Please add the information about their market weight.

Authors: addressed accordingly (over 160kg b.w.).

L173: Please add the space after 0.

Authors: addressed accordingly.

L225: Is it possible to provide readers with the photographs?

Authors: we are sorry, but we are unfortunately not able to include photographs.

L255: Are you sure about this citation and the value of 70 grams a day in that context? The paper you cited refers to the gastroprotective effects of straw. Should we automatically assume that the value represents the ideal daily straw needs?

Authors: we changed needs into consumption and clarified that these amounts were chosen because previous studies on this specific livestock production found they were actually consumed daily at weaning and fattening.

L384: Isn't is Authors'?

Authors: addressed accordingly.

L501: Please be consistent with the punctuation: e.g., (see L473).

Authors: addressed accordingly.

L709: 2022 (bold), 122 (not).

Authors: addressed accordingly.

L711: 2020 (bold).

Authors: addressed accordingly.

Reviewer 2 Report

Comments and Suggestions for Authors

Although the conclusions point out the need for a slow introduction of the tail docking ban, in intensive pig production factories. Showing that  not tail docking will cause economic loss and other  problems (e.g.  more use of drugs), strengthens the case for stopping tail docking.

The whole point of the EU legislation is to ensure that since pigs are recognised in law as sentient beings, there must be changes in intensive pig husbandry because such mutation has only been introduced to make more money. 

It is clear that changing one parameter, such as giving  them straw ( although it is not at all clear when they get given straw) is NOT going to change frustrated behaviours which are  the result of a whole collection of factors:, the substrate, the space, mixing groups, groups size, ages and sexes, lack of cognitive stimulus  etc etc etc. All of these  affect whether or not the pigs suffer prolonged frustration become aggressive, behave abnormally and show signs of distress.  Cutting off their tails is a  similar development ( now banned)  to the veterinarians cutting the muscles in the neck of horses to stop them performing stereotypes and would be the same as cutting of the thumbs of children  (other sentient  beings) because they suck them!

The  pigs have no freedoms to perform most of their natural behaviours so they are frustrated, and among other behavioural changes, aggression increases.  Unless many aspects of the environment are changed , this abnormal behaviour will not disappear. the ban on cutting of tails is a mutilation necessary to ensure economic gain when the pigs live  in inappropriate conditions, physical, socially emotionally and mentally.   but this can be radically changed by subsidises as is very well known. When the practise is banned, farmers will have to change  many aspects of the pigs environment to remain economic as is evident from this paper. These issues need to be discussed for this paper to have  value.   

Comments on the Quality of English Language

It is unnecessarily long winded particularly concerning the statistics which since the sample size of number of farms is not very great is not necessary, and no discussion of the ethics which is unacceptable.

Author Response

Comment 1: It is clear that changing one parameter, such as giving  them straw ( although it is not at all clear when they get given straw) is NOT going to change frustrated behaviours which are  the result of a whole collection of factors:, the substrate, the space, mixing groups, groups size, ages and sexes, lack of cognitive stimulus  etc etc etc. All of these  affect whether or not the pigs suffer prolonged frustration become aggressive, behave abnormally and show signs of distress.  Cutting off their tails is a  similar development ( now banned)  to the veterinarians cutting the muscles in the neck of horses to stop them performing stereotypes and would be the same as cutting of the thumbs of children  (other sentient  beings) because they suck them! The  pigs have no freedoms to perform most of their natural behaviours so they are frustrated, and among other behavioural changes, aggression increases.  Unless many aspects of the environment are changed , this abnormal behaviour will not disappear. the ban on cutting of tails is a mutilation necessary to ensure economic gain when the pigs live  in inappropriate conditions, physical, socially emotionally and mentally.   but this can be radically changed by subsidises as is very well known. When the practise is banned, farmers will have to change  many aspects of the pigs environment to remain economic as is evident from this paper. These issues need to be discussed for this paper to have  value. It is unnecessarily long winded particularly concerning the statistics which since the sample size of number of farms is not very great is not necessary, and no discussion of the ethics which is unacceptable.

Response 1: we have shortened some parts and discussed a bit more the ethics implications including the reviewer’s comments in the Discussion and Conclusions.

Reviewer 3 Report

Comments and Suggestions for Authors

Three phases of undocking pigs were examined in commercial farms related to economic effects. 

M&M

L182-186 In which way is behaviour monitored>

L278 Missing the effect of the farm/farmer/management. Were the results from some farms better than others. May be related to management differences. (See also L394, 480).

Author Response

Comment 1: L182-186 In which way is behaviour monitored>

Response 1: we have clarified that is is part of routine daily inspections, compulsory in accordance with Dir. 1998/58/EC.

Comment 2: L278 Missing the effect of the farm/farmer/management. Were the results from some farms better than others. May be related to management differences. (See also L394, 480).

Response 2: Acknowledging the potential influence of farm-specific variability, which could introduce collinearity given the unbalanced and non-random distribution of practices across farms, this initial exploratory analysis focused on estimating average practice effects without incorporating a farm-level random effect to avoid confounding practice effects. While we recognize that this approach does not account for potential additional variance attributable to farm, we anticipate that the size and direction of the observed average practice effects would likely remain consistent even with the inclusion of such a random effect, suggesting that our initial findings regarding the primary relationships are robust and warrant further investigation in subsequent, more complex modelling efforts.

Reviewer 4 Report

Comments and Suggestions for Authors

This research has significance for promoting animal welfare. For producers, especially economic input, is one of the important indicators to determine policies.  through the research data ,We can tailor policies to our national circumstances.

Comments on the Quality of English Language

1.Carefully revise the language in the article, a lot of the language is confusing

2.Detailed description of the actual implementation of each farm.

Author Response

Comment 1: This research has significance for promoting animal welfare. For producers, especially economic input, is one of the important indicators to determine policies.  through the research data ,We can tailor policies to our national circumstances.

Response 1: we thank the reviewer for the kind comment.

Comment 2: Carefully revise the language in the article, a lot of the language is confusing

Response 2: The English language has been revised as suggested.

Comment 3: Detailed description of the actual implementation of each farm.

Response 3: We added further information on the plan's implementation as supplementary material (Table S2). Moreover, we have added a sentence about the farm veterinarian verifying the actual implementation.

Round 2

Reviewer 2 Report

Comments and Suggestions for Authors

Yes this is better now some ethical concerns are mentioned and hopefully will be read. and it points out how many changes to the environment may be needed to cater for sentience in pigs.